# Preparation and Characterization of TiO_2_-Coated Hollow Glass Beads for Functionalization of Deproteinized Natural Rubber Latex via UVA-Activated Photocatalytic Degradation

**DOI:** 10.3390/polym15193885

**Published:** 2023-09-26

**Authors:** Supinya Nijpanich, Adun Nimpaiboon, Porntip Rojruthai, Jae-Hyeok Park, Takeshi Hagio, Ryoichi Ichino, Jitladda Sakdapipanich

**Affiliations:** 1Department of Chemistry, Center of Excellence for Innovation in Chemistry, Faculty of Science, Mahidol University, Bangkok 10400, Thailand; supinya@slri.or.th; 2Rubber Technology Research Centre, Faculty of Science, Mahidol University, Nakhon Pathom 73170, Thailand; adun.nim@mahidol.ac.th; 3Division of Chemical Industrial Process and Environment, Faculty of Science, Energy and Environment, King Mongkut’s University of Technology North Bangkok, Rayong 21120, Thailand; porntip.r@sciee.kmutnb.ac.th; 4Institute of Materials Innovation, Institutes of Innovation for Future Society, Nagoya University, Nagoya 464-8601, Japan; park.jae.hyeok.w3@f.mail.nagoya-u.ac.jp (J.-H.P.); hagio@mirai.nagoya-u.ac.jp (T.H.); ichino.ryoichi@material.nagoya-u.ac.jp (R.I.); 5Department of Chemical Systems Engineering, Graduate School of Engineering, Nagoya University, Nagoya 464-8603, Japan

**Keywords:** natural rubber latex, low-molecular-weight natural rubber latex, photochemical degradation process, titanium dioxide

## Abstract

The photochemical degradation of natural rubber (NR) is a prevalent method used to modify its inherent properties. Natural rubber, predominantly derived from the *Hevea Brasiliensis* tree, exhibits an exceptionally high molecular weight (MW), often reaching a million daltons (Da). This high MW restricts its solubility in various solvents and its reactivity with polar compounds, thereby constraining its versatile applications. In our previous work, we employed TiO_2_ in its powdered form as a photocatalyst for the functionalization of NR latex. However, the post-process separation and reuse of this powder present substantial challenges. In this present study, we aimed to functionalize deproteinized NR (DPNR) latex. We systematically reduced its MW via photochemical degradation under UVA irradiation facilitated by H_2_O_2_. To enhance the efficiency of the degradation process, we introduced TiO_2_-coated hollow glass beads (TiO_2_-HGBs) as photocatalysts. This approach offers the advantage of easy collection and repeated reuse. The modified DPNR showed a reduction in its number-average MW from 9.48 × 10^5^ to 0.28 × 10^5^ Da and incorporated functional groups, including hydroxyl, carbonyl, and epoxide. Remarkably, the TiO_2_-HGBs maintained their performance over seven cycles of reuse. Due to their superior efficacy, TiO_2_-HGBs stand out as promising photocatalysts for the advanced functionalization of NR across various practical applications.

## 1. Introduction

Natural rubber (NR) is a hydrocarbon polymer harvested from the *Hevea brasiliensis* tree. It comprises 94% *cis*-1,4-polyisoprene and 6% non-rubber components such as lipids, proteins, and inorganic constituents [1]. Due to its high molecular weight, NR exhibits excellent physical properties in resilience, strength, and fatigue resistance, making it desirable for various applications. However, NR is composed of hydrocarbons with very high molecular weights; therefore, it is difficult to process and obtain uniform composites due to its high viscosity and incompatibility with polar fillers and other molecules. The modification of NR latex, which involves reducing its molecular weight and imparting reactive functional groups, has attracted attention as a potential alternative for overcoming these disadvantages. Degraded and shorter NR chains have proven to be beneficial in a variety of applications, including as plasticizers [2], compatibilizers [3,4], adhesives [5], chain extension promoters [6,7,8], and grafting reaction linkers [9]. Approaches such as redox [10], oxidation [11], and photochemical degradation [12] can generate free radicals for the further preparation of low-molecular-weight NR (LNR). However, photochemical degradation is more promising than the alternatives because it is clean, low-energy, and non-toxic. Ravindran et al. [12] produced hydroxyl-terminated liquid NR in a rubber solution in the presence of hydrogen peroxide (H_2_O_2_) under ultraviolet (UV) irradiation from a medium-pressure mercury vapor lamp and sunlight.

It is well known that numerous semiconductors have been applied as photocatalysts, such as TiO_2_, CdS, ZnO, WO_3_, etc. [13,14,15,16]. ZnO and TiO_2_ have attracted attention as photocatalysts with wide applications due to their suitable morphologies, desired band gaps, and high surface areas, as well as their stability and reusability. However, ZnO has several drawbacks including an unsuitable band gap energy (*Eg* = 3.7 eV) and a low surface area or large particle size [17]. Therefore, TiO_2_ seems to be an interesting material with great properties, such as high stability, low cost, good biocompatibility, and environmental friendliness. It can be synthesized using a simple method.

Furthermore, our group is continuously studying the application of TiO_2_ photocatalyst powder, and we found that it shows good results for our purpose. However, the powdery form is impractical because it is difficult to collect and reuse after the process. Thus, we developed the TiO_2_-based photocatalyst in this work by coating hollow glass beads for facile collection and repetitive reuse.

TiO_2_ is a non-toxic and insoluble semiconductor powder used in various applications such as paints, cosmetics, and toothpaste [18]. Additionally, it is a well-known photocatalyst. TiO_2_ exists as three naturally occurring polymorphs with distinct characteristics: anatase, rutile, and brookite. Anatase permanently transforms into rutile at high temperatures. Due to a large band gap (3.23–3.59 eV) between the valence and conduction bands, the anatase form is recognized as the superior photocatalyst [19]. The photocatalytic activity of TiO_2_ is applicable in important technological areas [20], such as self-cleaning coatings [21,22], water splitting [23], water treatment [24,25], dye-sensitized solar cells [26,27], air purification [28], and self-sterilizing coatings [29]. The photocatalytic reaction on the TiO_2_ surface occurs when an electron (e^−^) in the valence band is excited by UV light (290–380 nm), thereby generating a positively charged hole (h^+^) in the conduction band. These excited h^+^ and e^−^ then transfer to the reactive species (i.e., O_2_ or H_2_O) adsorbed onto the catalyst surface, thus generating reactive free radicals that are crucial for the breakdown of organic molecules. A large surface area is essential for utilizing the photocatalytic capabilities of TiO_2_ effectively. Consequently, TiO_2_ powders for achieving high photocatalytic efficiencies have been explored. P-25 (Degussa) powder is a typical material with anatase and rutile phases with a considerably large surface area [30]. It is highly active in various photocatalytic processes, including destroying microcystin-LR, a cyanobacterial toxin in drinking water sources [31].

Due to its high photocatalytic efficiency, TiO_2_ has been used as a photocatalyst in the photodegradation of NR latex to achieve LNR. Fine TiO_2_ powder is advantageous for promoting the reaction; however, separating the powder after use is difficult. Our group previously employed TiO_2_-coated glass and quartz substrates as containers for NR latex to manufacture telechelic liquid NR [32,33]. The reaction was conducted for 5 h in the presence of H_2_O_2_ under UV irradiation. The glass substrate was reusable; however, the container size constrained the degradation process.

Moreover, functionalized styrene–butadiene rubber and skim NR latex were manufactured via photocatalysis using a TiO_2_-coated petri dish as the photocatalyst [34]. The hydroxyl group (-OH) amount increased with increasing UV-irradiation time but decreased after 5 h. In addition, a small amount of H_2_O_2_, 5% *w*/*w* of dry rubber, was sufficient for the photocatalysis of skim latex.

Recently, TiO_2_-coated Pyrex glass beads were employed as a photocatalyst to suppress algal growth in eutrophic water under UV illumination [35]. These beads considerably accelerated photocatalysis. In addition, they were easily isolated from the algae.

Therefore, in this study, TiO_2_-coated hollow borosilicate glass beads (TiO_2_-HGBs) were fabricated and utilized as photocatalysts to produce functionalized LNR (FLNR) latex in the presence of H_2_O_2_ under UV irradiation. The TiO_2_-HGBs were proposed to facilitate separation after usage by flotation and promote UV contact on the TiO_2_ surface during the reaction, thereby increasing the photocatalytic activity. Typically, an NR particle is surrounded by a layer of mixed protein and phospholipid domains [36]; the polyisoprene molecules subsequently form a hydrophobic core, resulting in a core–shell-like NR particle, as shown in Figure 1. The mixed layer may obscure NR particles making it difficult for them to react. This study used deproteinized NR (DPNR) latex as the starting material.

## 2. Materials and Methods

### 2.1. Materials

Fresh NR latex was generously provided by Thai Rubber Latex Corporation (Bangphli, Samutprakan, Thailand) Public Company Limited. Hollow borosilicate glass beads (HGBs) with an average diameter of 7 mm were made by the glassmaker shop. H_2_O_2_ (30% *w*/*w*), urea, and sodium dodecyl sulfate (SDS) were purchased from British Drug Houses (BDH Chemicals). Tetrapropyl orthotitanate was purchased from Fluka. Toluene, methanol, acetone, and tetrahydrofuran were purchased from RCI Labscan Limited. All chemicals were used without further purification.

### 2.2. Preparation of TiO_2_-Coated HGBs

First, a mixture of tetrapropyl orthotitanate (9.0 mL), distilled water (100.0 mL), and 70% *w*/*w* HNO_3_ (1.0 mL) was stirred for 24 h at 25 °C. After further stirring of the slurry for 6 h at 55 °C, a milky-looking TiO_2_ precursor sol was obtained.

Second, the HGBs were washed in acetone via ultrasonication for 1 h. The dried HGBs were then submerged in the synthesized TiO_2_ precursor sol, followed by dehydration and calcination at 550 °C for 1 h at a rate of 5 °C/min to eliminate the solvent and crystallize the TiO_2_ precursor layer.

### 2.3. Characterization of TiO_2_-HGBs

X-ray diffractometry with Cu-Kα1 radiation (λ = 0.15405980 nm) over the range of 20° < 2θ < 60° was used to analyze the crystallinity of the TiO_2_ films deposited on the HGBs. The step and time sizes were 0.02° and 30 s, respectively. We determined the mass percentages of the anatase and rutile phases using the following formulae, which are based on the intensity of the peak at 2θ values of 25.3° (101-plane) for anatase and 27.4° (110-plane) for rutile, respectively [37,38].
A (mass %) = 100/[1 + 1.265(I_R_/I_A_)](1)
R (mass %) = 100 − A = 100 × 1.265 I_R_/[I_A_ + 1.265 I_R_](2)

Here, A and R are the mass percentages of the anatase and rutile phases, respectively; I_A_ and I_R_ are the intensities of the (101) anatase and (110) rutile peaks.

Scanning electron microscope (SEM, JSM5410LV, JEOL^®^, Tokyo, Japan) equipped with energy dispersive X-ray spectrometer (EDS, Link ISIS-300, Oxford^®^, Abingdon, UK) was employed to analyze the surface morphologies of the HGBs before and after the formation of the TiO_2_ coating, and to determine the thickness of the TiO_2_ layer. Palladium sputtering was used in advance to impart conductivity to the samples, and further analysis was carried out by operating at an accelerating voltage of 20 kV under high vacuum mode while maintaining a working distance of 20 mm.

### 2.4. Preparation of FLNR Latex

In this study, DPNR latex was preliminarily prepared as the starting NR latex by incubating fresh NR latex (preserved with 0.3% *v*/*v* NH_3_) with 0.1% *w*/*v* urea and 1% *w*/*v* SDS at RT for 2 h while stirring. Then, the treated latex was centrifuged at 13,000 rpm and 25 °C for 30 min. The cream fraction was redispersed in 1% *w*/*v* SDS solution. After a second centrifugation under the same conditions, the cream fraction was redispersed in distilled water to obtain DPNR latex with a dry rubber content (DRC) of 30% [39].

A mixture of DPNR latex, H_2_O_2_, and TiO_2_-HGBs in a round-bottom flask was exposed to a 1000 W UVA lamp in a chamber. After irradiation, the FLNR was coagulated with acetone and dried at 40 °C under vacuum until a constant weight was reached. This procedure is the standard method used for preparing FLNR in subsequent investigations. The impact of H_2_O_2_ concentration in units of parts per hundred rubber (phr), TiO_2_-HGB quantity (number of beads), UVA-irradiation time, and the reusability of TiO_2_-HGBs were examined.

### 2.5. Characterization of FLNR Latex

FLNR and DPNR were re-precipitated in toluene/methanol before further analysis. The deproteinization was confirmed by investigating the nitrogen content of DPNR using a nitrogen analyzer (LECO FP-258) based on the combustion of oxygen gas. About 0.25 g of rubber sample was weighed and subjected to a nitrogen analyzer. The combustion of the rubber sample changed the nitrogen in the compounds to nitrogen gas which was detected as nitrogen content (% *w*/*w*). The results were obtained from the triplicate analysis. The weight-average molecular weight ($\bar{M_{w}}$*)* and number-average molecular weight ($\bar{M_{n}})$ of the prepared samples were determined using gel permeation chromatography (GPC−Waters^®^ 2414, Milford, CT, USA) using polystyrene standards and a refractive index (RI) detector equipped with a Styragel HR5E 7.8 × 300 mm column. The column was eluted with tetrahydrofuran (THF) at a 1.0 mL/min flow rate at 40 °C. The functional groups were investigated via Fourier-transform infrared spectroscopy (FTIR, FTIR-460 Plus, JASCO^®^, Tokyo, Japan) at RT at a resolution of 4 cm^−1^ and 16 scans per spectrum. The chemical microstructures were analyzed using ^1^H- and ^13^C-Nuclear Magnetic Resonance (NMR) spectroscopy at 50 °C using chloroform-*d* as the solvent. The measured frequency, pulse delay time, and number of scans for ^1^H- and ^13^C-NMR were 500 and 125 MHz, 4 and 5 s, and 500 and 12,000 scans, respectively.

Additionally, the gel contents of the samples were examined for side reactions by immersing the dried samples in toluene and storing them in the dark at RT for 7 days. The gel fraction was recovered by centrifugation, coagulated with methanol, and dried in an air-flow oven at 70 °C. The gel contents of the samples were estimated using the following formula:(3)$$Gel\;content\;\left(\%\right)=\frac{Weight\;of\;dried\;gel\;fraction\;of\;sample}{Weight\;of\;dried\;sample}\times 100$$

## 3. Results

### 3.1. Characterization of TiO_2_-Coated HGBs

The XRD pattern exhibits characteristic reflections of the anatase and rutile phases (Figure 2). The anatase phase of TiO_2_ shows diffraction peaks at approximately 25.3°, 37.8°, 38.5°, and 48.0°, corresponding to the (101), (004), (112), and (200) reflections, respectively. In addition, the rutile phase is identified by the peaks at 27.2°, 36.0°, 41.2°, and 54.3°, which are assigned to (110), (101), (111), and (211), respectively. Using Equations (1) and (2), the phase composition was estimated to be 68% anatase and 32% rutile. The mixed crystalline phases may have resulted from some anatase transitioning to the rutile phase during calcination. Recent reports show that the mixed phase of TiO_2_ (i.e., anatase and rutile) exhibits higher photocatalytic activity than either pure phase alone [40]. This result implies that the current procedure is suitable for synthesizing TiO_2_-HGBs to yield a mixture of anatase and rutile phases that may provide effective photoactivity for the photochemical degradation of organic compounds.

The morphology of the TiO_2_-HGBs was determined using SEM. The HGB surface is transparent and smooth before TiO_2_ coating (Figure 3a). Meanwhile, a uniformly dispersed porous TiO_2_ film is present on the HGB surface after TiO_2_ coating (Figure 3b), which is consistent with the report by Kim et al. [35]. These porous structures may have originated from the thermal decomposition of the components within the TiO_2_ precursor during calcination, which increases the porosity and specific surface area, causing the TiO_2_ film to exhibit significant photocatalytic efficiency [41]. SEM was also used to observe the cross-section of the TiO_2_-HGBs, and EDS analysis was performed. The thickness of the deposited TiO_2_ film is 297 nm (Figure 3c). The SEM-EDS analysis of Ti within the TiO_2_-HGBs using the mapping mode indicates the presence of a TiO_2_ layer on the HGB surface (Figure 3d). Additionally, the EDS spectrum of the TiO_2_-HGBs presented in Figure 3e confirms the successful coating of TiO_2_. It exhibits three additional Ti peaks in addition to the signals of oxygen, sodium, aluminum, and silicon, which are the primary components of the HGBs.

### 3.2. Preparation of FLNR Latex

#### 3.2.1. Effect of H_2_O_2_ Concentration

Ravindran T. et al. [12] discovered that H_2_O_2_ was essential for the formation of the hydroxyl radical (^•^OH). Herein, the optimal concentration of H_2_O_2_ was investigated, and GPC analysis was used to evaluate the degradation of the DPNR latex based on the reduction of its molecular weight. Figure 4a,b show the $\bar{M_{w}}$ and $\bar{M_{n}}$ of the product obtained using 20 g of DPNR latex at 10% DRC irradiated for 0 to 40 min under a 1000 W UVA lamp. The $\bar{M_{w}}$ and $\bar{M_{n}}$ of the samples gradually decrease as the H_2_O_2_ concentration increases, indicating that the ^•^OH produced by the dissociation of H_2_O_2_ under UVA irradiation plays an important role in DPNR degradation. Moreover, the $\bar{M_{w}}$ and $\bar{M_{n}}$ also decrease with increasing irradiation time up to 20 min, after which they are unaffected by exposure time up to 30 min. During this irradiation period, the $\bar{M_{w}}$ and $\bar{M_{n}}$ of the samples treated with 30 phr of H_2_O_2_ are comparable to those treated with 40 phr of H_2_O_2_. Therefore, the H_2_O_2_ concentration of 30 phr was considered adequate for breaking NR chains from DPNR latex.

#### 3.2.2. Effect of TiO_2_-HGB Quantity

The impact of TiO_2_-HGB quantity was investigated by measuring the molecular weight of the FLNR produced using varying amounts of TiO_2_-HGBs (0–35 pieces) in the presence of 30 phr of H_2_O_2_. The reaction was conducted for 30 min under UVA irradiation. The $\bar{M_{w}}$ and $\bar{M_{n}}$ of the FLNR samples decreased with the increasing amount of TiO_2_-HGBs and reached a minimum when 25 pieces of TiO_2_-HGB were utilized (equal to 25 mmol of TiO_2_ calculated using the difference in HGB weight before and after TiO_2_ coating) (Figure 5a). Therefore, 25 pieces of TiO_2_-HGB were regarded as the optimal quantity for FLNR preparation, obtaining a $\bar{M_{w}}$ and $\bar{M_{n}}$ of 1.43 × 10^5^ and 0.28 × 10^5^ Da, respectively. However, when 30 and 35 pieces of TiO_2_-HGB were employed, the $\bar{M_{w}}$ and $\bar{M_{n}}$ of the FLNR increased. This trend was expected because excess HGBs may disable the UVA light from penetrating the photochemical degradation cycle.

Under UV irradiation, cross-linking normally occurs as a competitive reaction in the presence of reactive radical species, that is, ^•^OH. The gel content of the FLNR samples was also assessed to explore cross-links during the photochemical degradation process. Figure 5b shows the gel content of the samples with varying amounts of TiO_2_-HGBs. NR gel typically results from the interactions between non-rubber components (i.e., proteins and phospholipids) [42]. The gel content dropped from 5.31 to 1.35% *w*/*w* with the addition of H_2_O_2_ under UVA irradiation. This reduction in gel content suggests that the gel fraction of DPNR degrades owing to the chain scission generated by photooxidation in the H_2_O_2_/UVA system [43]. The gel fraction decreased to approximately 0.32% *w*/*w* with the addition of 5 pieces of TiO_2_-HGB and became almost zero with the addition of 10 pieces of TiO_2_-HGB. Therefore, an increase in TiO_2_-HGB quantity might expedite the chain scission of rubber in DPNR latex without any cross-linking reaction being detected.

Figure 6 illustrates the FTIR spectra of DPNR (control) and FLNR produced using varying amounts of TiO_2_-HGBs. For the DPNR spectrum, the peaks at 3280 cm^−1^ and 1548 cm^−1^, characteristic bands of NR proteins, disappeared [44]. Moreover, the nitrogen content of DPNR was also determined and found to be only 0.04 % *w*/*w*. These results confirm the reduction in proteins after the deproteinization of NR latex, which is possibly due to the weak attractive forces between proteins and the NR particle surface. After centrifugation, most proteins detach themselves from the rubber particles by forming hydrogen bonds with urea.

In the comparison of the FTIR spectra of DPNR and FLNR prepared using various TiO_2_-HGB quantities, all spectra reveal the distinctive characteristic peaks of NR at 1664, 1375, and 836 cm^−1^, which are attributed to C=C stretching, –CH_3_ asymmetric, and =C–H deformations, respectively. New broad bands corresponding to the hydroxyl (-OH), carbonyl (C=O), and epoxide groups were observed in the FLNR spectrum at 3450, 1716, and 873 cm^−1^, respectively. Interestingly, the intensity of the –OH and C=O groups increased as the amount of TiO_2_-HGBs increased to 25 pieces. This result indicates that a larger quantity of HGBs promotes a more active site on the TiO_2_ film, thereby increasing ^•^OH generation. Thus, DPNR particles had a high opportunity of being attacked by ^•^OH, leading to their depolymerization and functionalization. However, the intensities decreased when more than 25 pieces of TiO_2_-HGB were utilized. It was hypothesized that the high concentration of TiO_2_-HGBs might absorb and conceal the UV light, resulting in reduced energy interaction between the TiO_2_-HGBs, rubber particles, and H_2_O_2_ molecules. These results aligned with those of the molecular weight results.

#### 3.2.3. Effect of UVA-Irradiation Time

Isa et al. reported that temperature and reaction time can influence the degradation of NR latex [45]. Here, the effect of the irradiation period was evaluated by observing the decrease in the $\bar{M_{w}}$ and $\bar{M_{n}}$ of the FLNR samples (Figure 7a). After 30 min of irradiation, the $\bar{M_{w}}$ decreased from 1.13 × 10^6^ to 1.48 × 10^5^ Da similar to the $\bar{M_{n}}$ that decreased from 9.48 × 10^5^ to 0.34 × 10^5^ Da. A longer irradiation period facilitated the degradation of more NR chains into shorter segments. However, a prolonged reaction time can accelerate radical formation, leading to cross-linking. Figure 7b shows the gel content of the FLNR samples after various irradiation periods. The gel content decreases considerably after 10 min of irradiation. It disappears almost completely after 30 min, suggesting that no cross-linking occurs after 30 min of UVA exposure.

Figure 8 shows the FTIR spectra of the DPNR (control) and FLNR samples. The intensities of the –OH and C=O groups increased steadily with increasing exposure duration, reaching their maximum ratios after 30 min of UVA irradiation. Nevertheless, the –OH and C=O contents decreased after 40 min of exposure. This result may be because a long irradiation time results in a high production rate of ^•^OH, which can oxidize the functional groups on the NR chains and form free radicals in the system.

Moreover, the structure of the produced samples was analyzed using ^1^H- and ^13^C-NMR spectroscopies. Figure 9 shows the ^1^H-NMR spectra of DPNR and FLNR. The characteristic signals of *cis*-1,4-polyisoprene units are detected in both samples at 1.70, 2.07, and 5.15 ppm, corresponding to the methyl (–CH_3_), methylene (–CH_2_–), and methine (=CH–) protons, respectively [10]. Additional signals between the chemical shifts of 3.4 and 4.2 ppm assigned to the –OH groups on the NR chains were detected in FLNR (Figure 9b) [46]. In addition, signals at 2.71 and 1.28 ppm suggest the presence of methylenic and methylic protons in the epoxide ring, respectively. Furthermore, the presence of C=O groups is indicated by an additional signal at 2.20 ppm and two small signals at 9.41 and 9.83 ppm. The signal at 2.20 ppm corresponds to the protons on the carbon adjacent to the C=O. However, the signals at 9.41 and 9.83 ppm are assigned to the aldehydic protons that have shifted far downfield due to the anisotropy of the C=O groups.

The ^13^C-NMR spectra of the purified DPNR and FLNR are shown in Figure 10. The signals at 23.37, 26.40, 32.21, 125.03, and 135.17 ppm are the characteristic signals of NR. The small signal at 68.17 ppm corresponding to the carbon attached to the –OH groups was detected in FLNR (Figure 10b). Additional signals at 60.75 and 64.49 in the FLNR spectrum correspond to the carbon of the epoxide ring [47]. A carbon signal close to that of C=O was observed at 38.81. The thermal degradation of NR can form C=O and epoxide groups once a high temperature is reached during irradiation.

These results are consistent with the molecular weight results that suggest that the breakage of the NR chain results in a decrease in the molecular weight and the introduction of a functional group at the point of chain scission. The proposed mechanism is illustrated in Figure 11.

When ^•^OH is generated via the dissociation of H_2_O_2_ under UV irradiation, it attacks the broken single bond and forms a hydroxyl-terminated chain (Path I). In another case, the double bond of the *cis*-1,4-polyisoprene chain breaks first, and then the ^•^OH attacks the broken points, forming a hydroxyl group that transforms into an epoxide (Path II). However, the epoxide group can be hydrolyzed, forming hydroxyl groups attached to the chain.

### 3.3. Reusability of TiO_2_-HGBs

Reusability was a major advantage of the TiO_2_-HGBs used in the current study. The effectiveness of the TiO_2_-HGBs was measured by reusing them nine times. Figure 12I presents the SEM images of the TiO_2_-HGB surface and the cross-sectional SEM images of the TiO_2_-HGBs after the photochemical degradation of DPNR latex. After using the TiO_2_-HGBs four times (Figure 12(Ib)), the TiO_2_ layer and some components suspected to be degraded NR are still visible on the HGB surface. However, the TiO_2_ film thins out and is damaged during photochemical degradation, as confirmed by the SEM images of the cross-section in Figure 12(Ie). The thickness of the TiO_2_ film decreases from approximately 297 to 246 nm. After nine uses, the TiO_2_ film is covered by layers that could be leftover NR (Figure 12(Ic)). The TiO_2_ film becomes thinner, to approximately 111 nm, as shown in Figure 12(If).

Figure 12II shows the elemental composition obtained from the EDS analysis of the TiO_2_-HGB surface. The amounts of C and Si increase slightly after using FLNR nine times. The increased C content may be attributed to the numerous carbon atoms in the remaining NR. Contrastingly, the decrease in the quantity of Ti after four and nine uses corresponds to the reduced thickness of the TiO_2_ layer, as shown in the SEM images of the TiO_2_-HGB cross-section.

Figure 13 shows the $\bar{M_{w}}$ of FLNR samples at different recycling times. FLNR was prepared using the same 25 pieces of TiO_2_-HGB in the presence of 30 phr of H_2_O_2_ for 30 min under UVA irradiation. The $\bar{M_{w}}$ of the samples did not change much after using the TiO_2_-HGBs up to four times. After seven uses, the $\bar{M_{w}}$ slightly increases, but it is acceptable for FLNR preparation. However, after eight and nine applications of the TiO_2_-HGBs, the $\bar{M_{w}}$ of FLNR dramatically increases. The $\bar{M_{w}}$ of the FLNR samples increased to approximately 2.20 × 10^5^ Da after using the TiO_2_-HGBs nine times, suggesting that the effectiveness of the TiO_2_-HGBs weakened after seven cycles. From all evidence, despite the destruction of a few TiO_2_ films during the photochemical degradation, TiO_2_-HGBs can be reused with excellent performance at least seven times.

## 4. Conclusions

This study elucidates the preparation and comprehensive characterization of TiO_2_-coated hollow glass beads (TiO_2_-HGBs) employed to functionalize deproteinized natural rubber (DPNR) latex via photocatalytic degradation. This process was facilitated under UVA irradiation, utilizing H_2_O_2_ as the oxidizing agent. The TiO_2_ film, comprising anatase and rutile phases, manifested uniform coatings on the HGBs, with a thickness of approximating 297 nm. Upon exposure to 1000 W of UVA irradiation for 30 min, and in the simultaneous presence of 30 phr of H_2_O_2_ and 25 individual TiO_2_-HGBs, the number-average molecular weight ($\bar{M_{n}}$) of functionalized low-molecular-weight NR (FLNR) notably diminished to 0.28 × 10^5^ Da without any discernible gel formation. The truncated DPNR chains incorporated hydroxyl, carbonyl, and epoxide groups attributed to the ^•^OH radicals binding at their rupture points. Notably, the TiO_2_-HGBs demonstrated consistent efficacy over seven cycles of reuse. Given these findings, TiO_2_-HGBs are proposed as potent photocatalysts for the photochemical degradation of DPNR in latex form, underpinned by their distinguished physical and structural attributes, minimal toxicity, and robust chemical stability. This approach not only indicates potential for optimization, but also seems suitable for practical applications and potential commercialization.

## Figures and Tables

**Figure 1 polymers-15-03885-f001:**
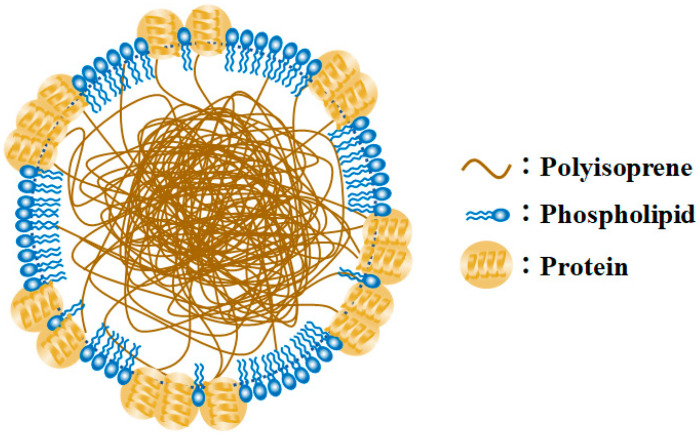
Model of latex particle surrounded by a mixture of proteins and phospholipids (Modified from [36]).

**Figure 2 polymers-15-03885-f002:**
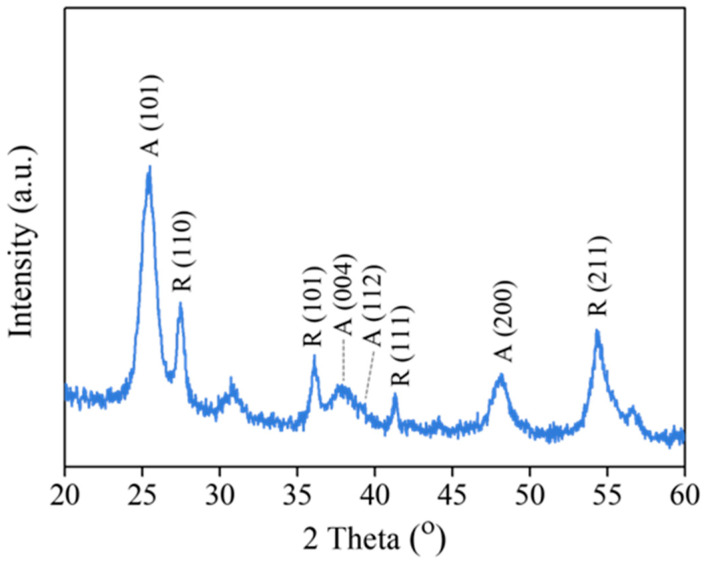
XRD pattern of TiO_2_ film coated on HGBs.

**Figure 3 polymers-15-03885-f003:**
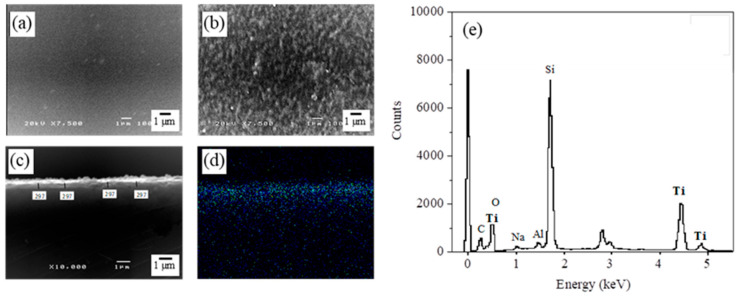
SEM images of the HGB surface (**a**) before and (**b**) after TiO_2_ coating at 7500X. The cross-sectional SEM images of the TiO_2_-HGBs in (**c**) secondary electron image (SEI) and (**d**) Ti element mapping. (**e**) The EDS spectrum of elements on the TiO_2_-HGBs.

**Figure 4 polymers-15-03885-f004:**
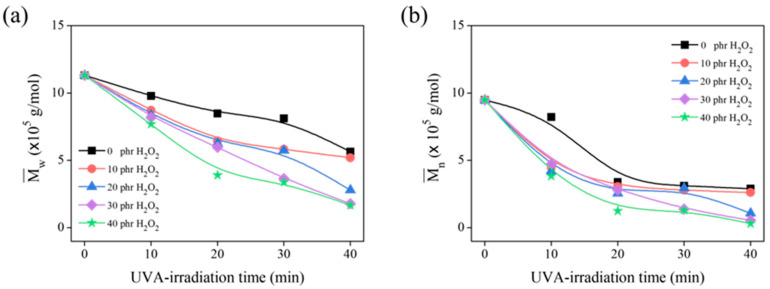
The (**a**) $\bar{M_{w}}$ and (**b**) $\bar{M_{n}}$ of DPNR latex in the presence of 0, 10, 20, 30, and 40 phr of H_2_O_2_ after several UVA-irradiation time intervals.

**Figure 5 polymers-15-03885-f005:**
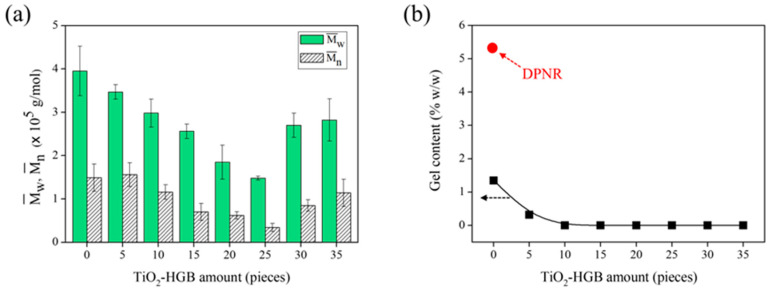
(**a**) The $\bar{M_{w}}$ and $\bar{M_{n}}$ and (**b**) gel content of DPNR and FLNR samples prepared via UVA irradiation for 30 min in the presence of 30 phr of H_2_O_2_ using various amounts of TiO_2_-HGBs.

**Figure 6 polymers-15-03885-f006:**
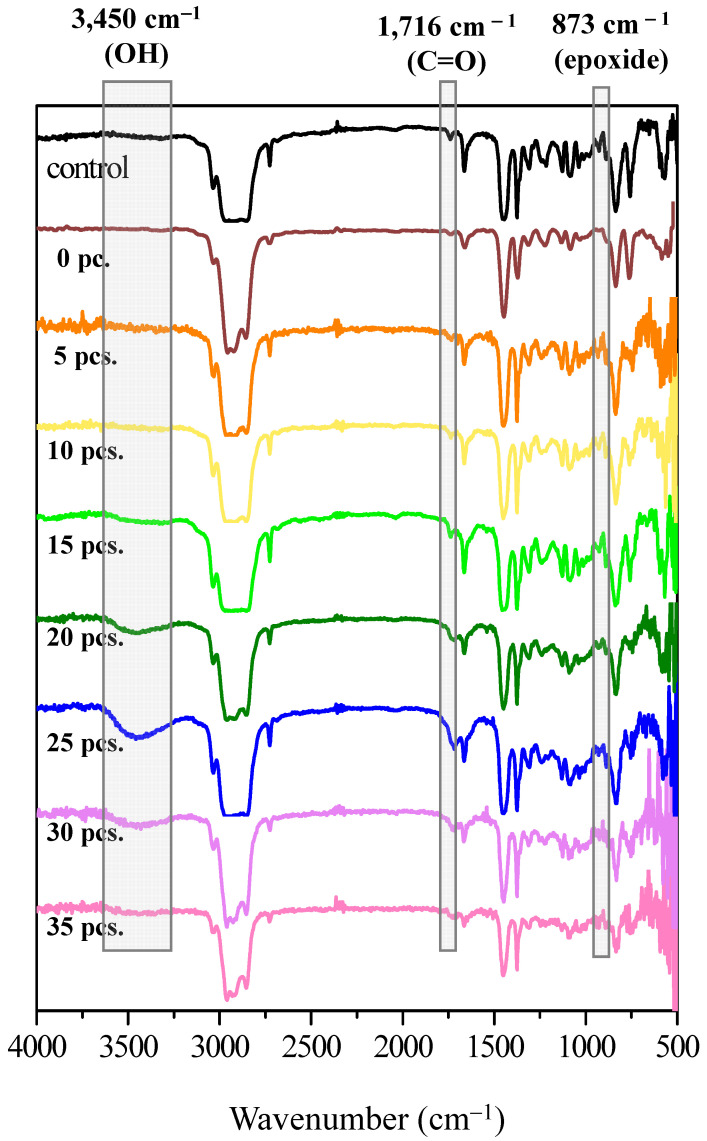
FTIR spectra of DPNR (0 piece) and FLNR samples prepared via UVA irradiation for 30 min in the presence of 30 phr of H_2_O_2_ using various quantities of TiO_2_-HGBs.

**Figure 7 polymers-15-03885-f007:**
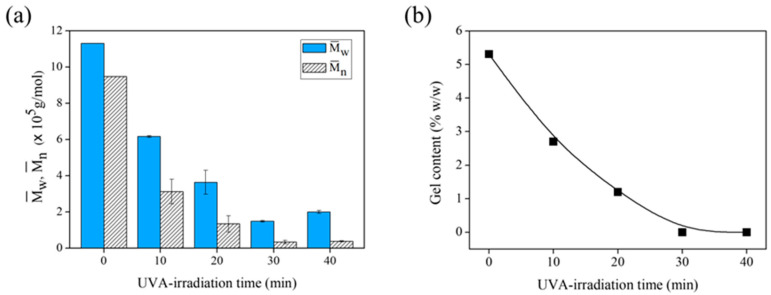
(**a**) The $\bar{M_{w}}\;$ and $\bar{M_{n}}$ and (**b**) gel content of DPNR (no TiO_2_-HGB) and FLNR samples prepared in the presence of 30 phr of H_2_O_2_ using 25 pieces of TiO_2_-HGB under different UVA-irradiation time intervals.

**Figure 8 polymers-15-03885-f008:**
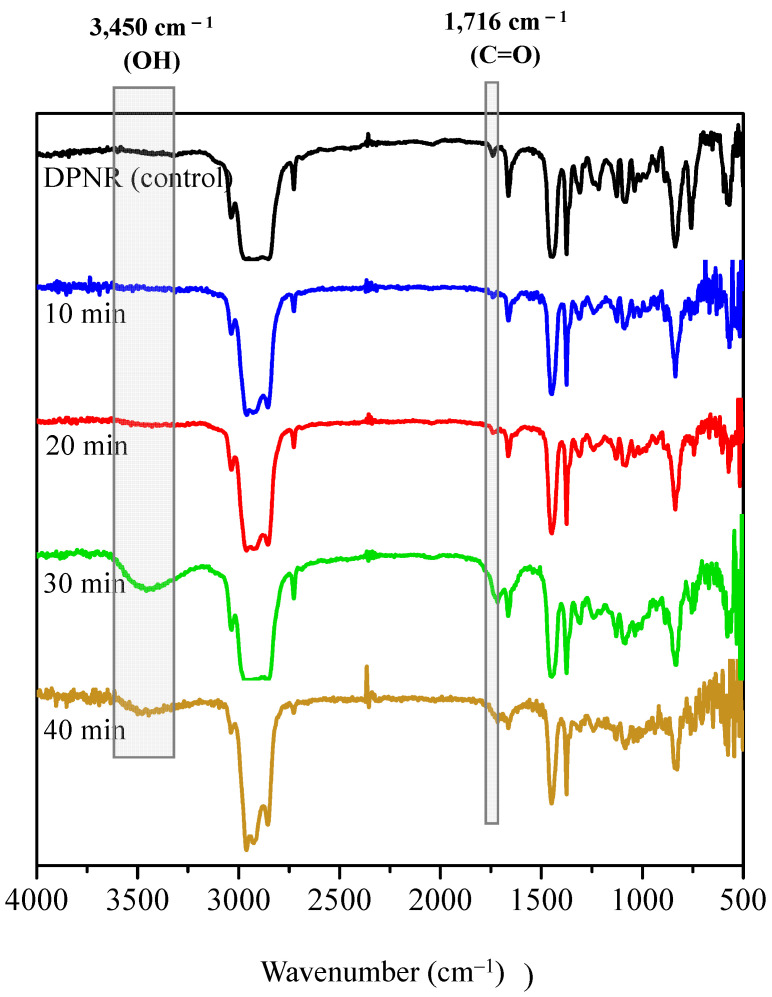
FTIR spectra of DPNR (control) and FLNR samples in the presence of 30 phr of H_2_O_2_ and 25 pieces of TiO_2_-HGB at various irradiation time intervals.

**Figure 9 polymers-15-03885-f009:**
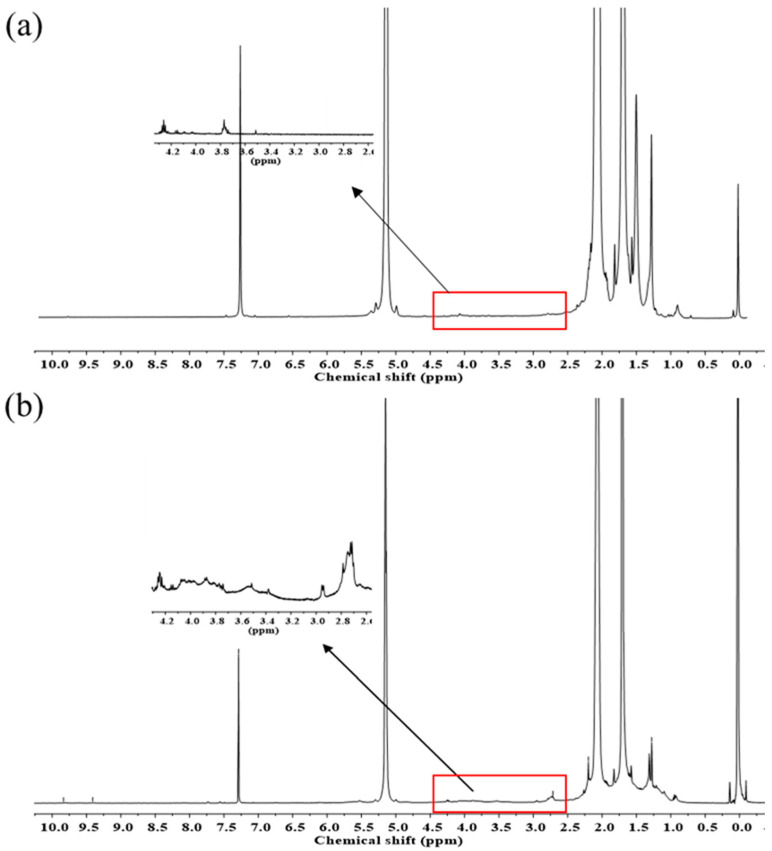
^1^H-NMR spectra of purified (**a**) DPNR and (**b**) FLNR.

**Figure 10 polymers-15-03885-f010:**
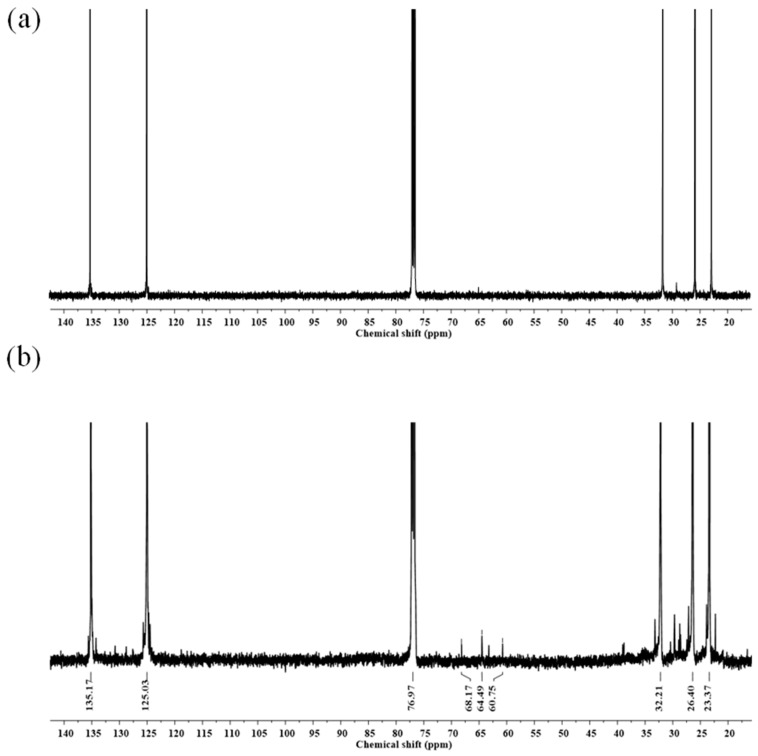
^13^C-NMR spectra of purified (**a**) DPNR and (**b**) FLNR.

**Figure 11 polymers-15-03885-f011:**
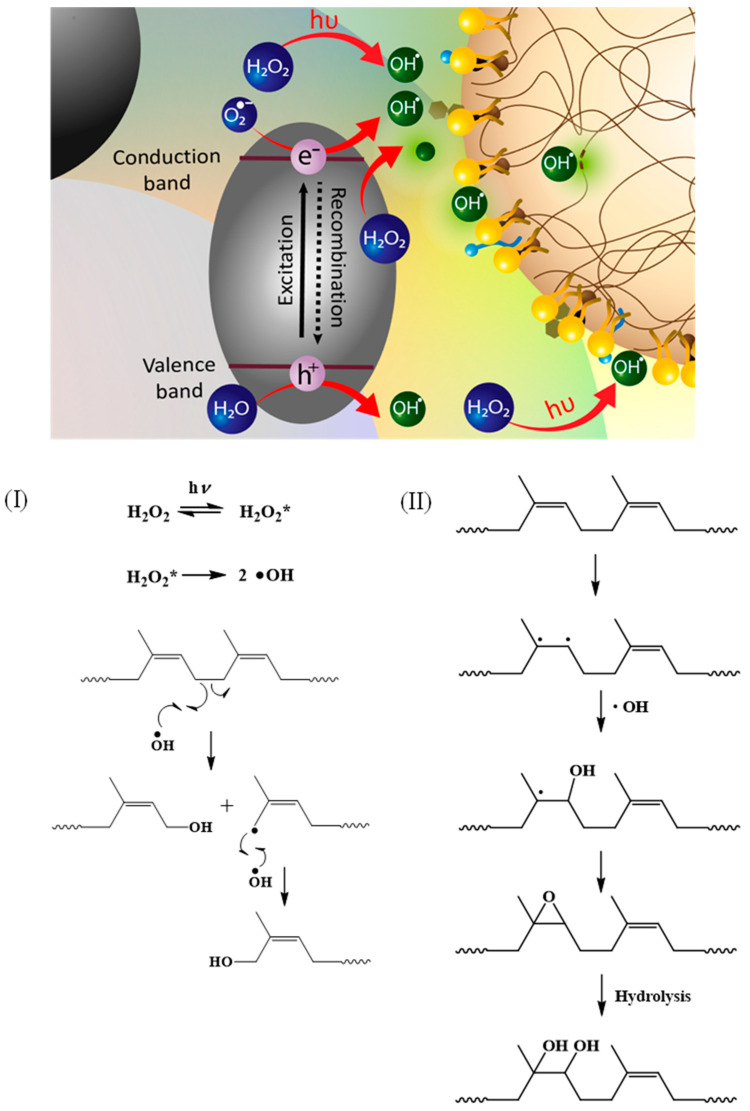
Possible proposed mechanisms of the photochemical degradation process of DPNR latex. Path (**I**): Reaction at single bond of DPNR chain and Path (**II**): reaction at double bond of DPNR chain.

**Figure 12 polymers-15-03885-f012:**
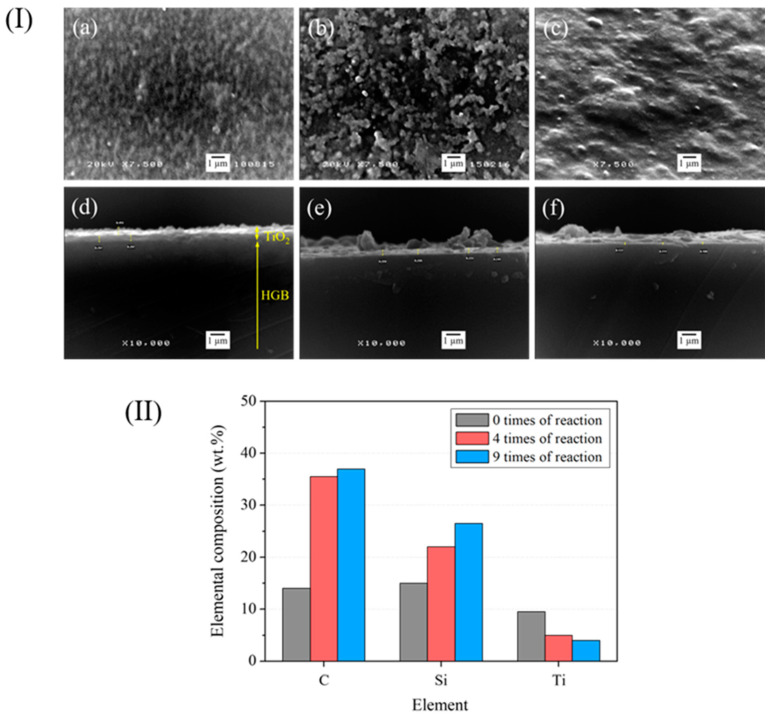
(**I**) SEM images of the TiO_2_-HGB surface (**a**–**c**) and the cross-section (**d**–**f**). (**II**) Quantity of elements on the TiO_2_-HGB surface after using for 0, 4, and 9 times, respectively.

**Figure 13 polymers-15-03885-f013:**
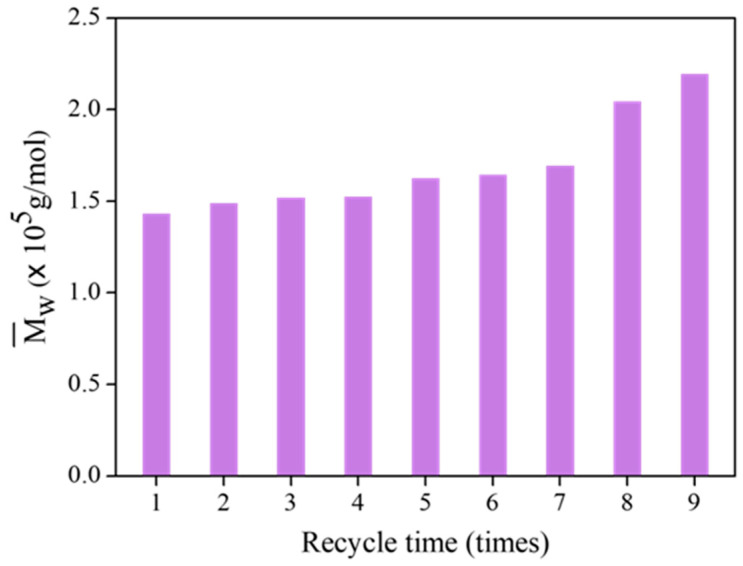
The $\bar{M_{w}}$ of FLNR samples at various recycling times.

## Data Availability

Not applicable.

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
