# Peer review of "Preparation and Characterization of TiO2-Coated Hollow Glass Beads for Functionalization of Deproteinized Natural Rubber Latex via UVA-Activated Photocatalytic Degradation"

_polymers, 2023, doi:10.3390/polym15193885_

Round 1

Reviewer 1 Report

1.                   The title is not scientific and capitalize the first alphabet of each word in the title and keywords section and so one.

2.                   Abstracts are too concise. Represent a clear comparison of your work with previous reported work in the abstract and conclusion section, show the novelty specially.

3.                   In the introduction portion put the whole name of abbreviation for formula such as H2O2 etc.

4.                   There is many photocatalysts that have good photocatalytic performance and latest based on 2D semiconductor, then why you used TiO2.? The photocatalyst you used is too old, employed too much and not active in visible region due to large band gap.

5.                   You discussed and given high surface area of this photocatalyst, please give references.

6.                   Fig. 1 is adopted or designed by you? If it is adopted, then give reference and copyright permission.

7.                   After further maturing the slurry for 6 h at 55˚C, a milky-looking TiO2 precursor sol was obtained. The words, specially maturing is not appropriate.

8.                   Draw the schematic Fig. 10 in high resolution with reaction steps and discuss in detail.

9.                   Paragraphs in several sections are extremely long and span several pages. The message would come across a lot more clearly with more concise and structured writing.

10.               Provide the TEM morphologies of as-prepared material with discussion.

11.               Provide the BET isotherms pore size graph with discussion. How much surface area is of your materials. Show the comparison before and after modification.

12.               You have shown only a single XRD of one material. The XRD after modification or before not provided.

13.               Diffuse reflectance spectroscopy (DRS) is important to be given for a photocatalyst and to show on such analysis the band gap and optical absorption.

14.               Conclusion section is too short to make it a bit comprehensive.  

15.               Journal names in the references section are inconsistently abbreviated.

16.               The manuscript requires some language improvements for grammatical and syntactical accuracy.

17.               Double check the English corrections, grammar checks, pronunciation, punctuation, and other essential changes in the revised manuscript copy.

The English corrections, grammar checks, pronunciation, punctuation, and other essential changes in the revised manuscript copy are important. 

Author Response

Dear Reviewer of POLYMERS Journal

We would like to appreciate for your kind consideration our manuscript (polymers-2551857) entitled “Fabrication of Functionalized Deproteinized Natural Rubber Latex Using TiO2-coated Hollow Glass Beads under UVA-driven Photocatalytic Degradation Process” by Supinya Nijpanich, Adun Nimpaiboon, Porntip Rojruthai, Jae-Hyeok Park, Takeshi Hagio, Ryoichi Ichino and Jitladda Sakdapipanich*. We would like to express our appreciation to you for the comments and helpful suggestions. We have revised the manuscript according to the comments of you.

            The responses and the details of correction are given below, and the corresponding revisions made to the manuscript has been marked up using the “Track Changes” function. In addition, we have improved three figures, i.e., adding a comparative spectrum in Figure 9 and 10 (the former Figure 9); 1H- and 13C-NMR spectra, respectively, and improving the resolution in Figure 11 (the former Figure 10). Please note that we have changed the manuscript’s title to “Preparation and Characterization of TiO-coated Hollow Glass Beads for Functionalization of Deproteinized Natural Rubber Latex via UVA-activated Photocatalytic Degradation

We would like to appreciate for your kind consideration again. We would be grateful if our manuscript could be accepted to publish in POLYMERS (Special Issue "Degradation and Stability of Polymer Based Systems").

Sincerely Yours,

Professor Dr. Jitladda Sakdapipanich

Reviewer 2 Report

The article is a different perspective of the published one of the authors from 2022, reference 27. 

Different aspects must be revised:

- the abstract presents a different perspective of the actual content of the article, the functionalization via a photochemical degradation process under UVA irradiation in the presence of H2O2 as an oxidizing agent, but it also presents the TiO2-coated hollow glass beads  synthesis and charcaterization, therefore, please revise it;

-  the introduction is a brief presentation of NR and some info about TIO2 beads, no data about the actual stage of research in this field; for example the motivation of the whole process;

- the novelty is the TiO2 glass beads? or the use as photocatalysts to produce functionalized LNR? it is not clear, please mention;

- how did the authors know that the process of DPNR latex was done? is there any confirmation that the deproteinization happen?

- also, the C-NMR is just 1 spectra, no a and b as the H-NMR, please add it.

- it is not clear how the TIO2 HGB look like? the presented SEM images are blurry, could not identify anything, please revise;

- the conclusion does not present the actual data from the article, just a small section is pointed out, please revise it;

- does the dimension of the glass bead influence the molecular weight of the FLNR?

Author Response

(The authors gave the same response as above.)

Round 2

Reviewer 2 Report

The authors resolved all the pointed issues. 

The text requires minor corrections by a native speaker.